# Fatty Liver Change in Korean Adults in a Systematic Social Distancing System Amid the COVID-19 Pandemic: A Multicenter Analysis

**DOI:** 10.3390/ijerph191610444

**Published:** 2022-08-22

**Authors:** Ji-Hee Haam, Yang-Im Hur, Young-Sang Kim, Kyoung-Kon Kim, Jee-Hyun Kang, Hae-Jin Ko, Yoon Jeong Cho, Hye-In Choi, Kyu Rae Lee, Jung Ha Park, Soo Hyun Cho, Jong-Koo Kim, Taesic Lee, Myung-Jae Seo, Yeong Sook Yoon, Yoobin Seo, Ga Eun Nam, Sun Hyun Kim

**Affiliations:** 1Department of Family Medicine, CHA Bundang Medical Center, CHA University, Seongnam 13496, Korea; 2Chaum Life Center, CHA University, Seoul 06062, Korea; 3Department of Family Medicine, Gachon University College of Medicine, Gachon University Gil Medical Center, Incheon 21565, Korea; 4Department of Family Medicine, Konyang University College of Medicine, Daejeon 35365, Korea; 5Department of Family Medicine, Kyungpook National University Hospital, Daegu 41944, Korea; 6Department of Family Medicine, Daegu Catholic University, Daegu 42472, Korea; 7Department of Family Medicine, Jeju National University Hospital, Jeju 63241, Korea; 8Department of Family Medicine, College of Medicine, Chung-Ang University Hospital, Seoul 06973, Korea; 9Department of Family Medicine, Yonsei University Wonju College of Medicine, Wonju 26426, Korea; 10Department of Family Medicine, Inje University Ilsan Paik Hospital, Goyang 10380, Korea; 11Department of Family Medicine, Sanbon Medical Center, Wonkwang University, Gunpo 15865, Korea; 12Department of Family Medicine, Korea University Guro Hospital, Seoul 08308, Korea; 13Department of Family Medicine, International St. Mary’s Hospital, Catholic Kwandong University College of Medicine, Incheon 22711, Korea

**Keywords:** COVID-19, systematic social distancing, fatty liver, liver enzyme, alcohol consumption

## Abstract

In response to the COVID-19 pandemic, the Korean government implemented policies including the systematic social distancing (SSD) system which started on 28 June 2020. The present study investigated the development and aggravation of fatty liver measured using ultrasonography during the transition period (from pre-SSD to SSD) compared to the fatty liver changes during the pre-SSD period. Changes in fatty liver and liver enzymes were assessed in different groups stratified by alcohol consumption. Our retrospective cohort analysis included 5668 participants who underwent health checkups at 13 university hospitals during the SSD period and two or more checkups before the SSD period. Fatty liver developed and aggravated more in the transition period (13.6% development and 12.0% aggravation) than in the pre-SSD period (10.8% development and 10.1% aggravation) in the alcohol consumption group. This finding was more prominent in women than in men. Abnormal alanine transaminase levels were more often developed in the transition period than in the pre-SSD period, especially in men (11.1% vs. 8.6% in each period). In conclusion, the SSD system may contribute to fatty liver changes in individuals who regularly consume alcohol. Further research of the post-pandemic period is needed to assess long-term changes in fatty liver disease.

## 1. Introduction

The COVID-19 pandemic has impacted all areas of life and has prompted significant changes in economic, environmental, and social domains [1,2]. Policies such as localized lockdowns, social distancing, and self-quarantine are influencing various dimensions of lifestyle and physical and psychological health [3,4]. Consequently, there have been population-level changes in health behaviors such as diet and physical activity with negative effects on obesity and obesity-linked metabolic disease [5,6,7,8,9].

Hepatic steatosis or fatty liver is defined clinically by intrahepatic fat of at least 5% of liver weight [10]. Alcohol consumption is known as a major risk factor for fatty liver disease. As alcohol and its metabolites have toxic effects on the liver, more consumption of alcohol contributes to a higher likelihood of development of alcoholic fatty liver disease [11,12]. Nonalcoholic fatty liver disease (NAFLD) is a hepatic steatosis without a secondary contributing factor such as excess alcohol intake, viral infection, or drug treatment [10,13]. The risk factors for NAFLD are physical inactivity, dyslipidemia, diabetes, and metabolic syndrome [14,15,16]. Thus, lifestyle modifications consisting of diet, exercise, and weight loss are recommended treatments for NAFLD [15].

The Korean government revised their COVID-19 policies many times during the pandemic. The first wave of COVID-19 with a large-scale infection rate occurred between February and March 2020 [17]. Subsequently, a more detailed and systematic social distancing (SSD) system was introduced on 28 June 2020, which mandated social distancing and included regulation of facility use and social gatherings [18]. These regulations influenced personal behaviors such as physical activity, diet, and alcohol consumption [19,20,21,22]. For example, studies reported increased alcohol consumption and decreased physical activity during the COVID-19 pandemic [23,24,25]. In addition, quarantine was associated with psychiatric instabilities such as depressive and anxiety symptoms [19,24]. It has been proposed that these changes may have contributed to a deterioration of metabolic parameters in some individuals [26,27,28]. These lifestyle changes may influence liver fat accumulation. However, it remains unknown if the epidemiologic status of hepatic steatosis changed during the pandemic. It could be hypothesized that SSD may influence the prevalence of fatty liver in the population under the regulation. In the current study, development and aggravation of fatty liver in the non-drinker and alcohol consumption groups were compared between the time intervals with and without the influence of SSD.

## 2. Materials and Methods

### 2.1. Study Design and Subjects

This retrospective cohort study was conducted based on the data from health checkups at university hospitals. Thirteen health checkup centers participated in this study: four in Seoul, two in Gyeonggi Province, two in Incheon, one in Daejeon, two in Daegu, one in Wonju, and one in Jeju.

The period from July 2020 (the beginning of the SSD) to the time immediately before the initiation of this study (June 2021) was defined as the SSD period. Among the subjects who visited each hospital for regular health checkup during the SSD period, those who visited two or more times between January 2018 and June 2020 were included in our study. Data from visits that included colonoscopy preparation were not included in the analysis to avoid the influence of excessive dehydration (Figure 1).

Regarding the visits, three assessment points were defined for each subject (Figure 2): the SSD period (visit 3), the time immediately before the SSD period (visit 2), and the time before visit 2 (visit 1, the baseline; the nearest time from the year before visit 2).

The exclusion criteria were as follows: the presence of thyroid disease, uncontrolled diabetes, suspected chronic kidney disease, positive anti-HIV antibodies, history of malignancies, suspected malignancies detected by endoscopy, abdominal ultrasonography, chest CT, or abdominopelvic CT, long-term use of medications related to fatty liver changes such as steroids, anti-obesity drugs, sulfonylurea, thiazolidinediones, insulin, antidepressants, antipsychotics, amiodarone, methotrexate, tamoxifen, valproate, phenobarbital, carbamazepine, antiviral agents, tetracycline, benzbromarone, etc. [29,30], and missing anthropometric data. A total of 8109 subjects were included in the initial dataset.

We further screened our initial pool of participants for the following criteria. Individuals with a history of infection or seropositivity for HBV and/or HCV and those without information on drinking habits were excluded. In addition, data from visits without abdominal ultrasonography were excluded. Additionally, individuals were only included in the final analysis if data were collected on them during the SSD period and two or more checkups before the SSD period. Application of these additional criteria left us with 5668 subjects for the final analysis.

### 2.2. Primary and Secondary Endpoints

The primary endpoints were development and aggravation of fatty liver assessed using ultrasonography in both the non-drinker and alcohol consumption groups. The secondary endpoints were loss of normality in alanine transaminase (ALT) and FIB-4 in each group.

### 2.3. Measurements and Personal Medical History

Personal medical histories were collected using self-reported questionnaires. The participants were classified according to smoking history as nonsmokers, ex-smokers, and current smokers. Those who answered “yes” regarding drinking were considered alcohol consumers.

Height and weight were measured in a standing position and recorded to the first decimal point in centimeters and kilograms, respectively. Body mass index (BMI) was calculated by dividing weight (kg) by the square of height (m^2^). Obesity was defined as the BMI of 25 kg/m^2^ or more. Blood pressure (BP) was analyzed using an automatic sphygmomanometer with an appropriate cuff size after resting for 10 min in the sitting position.

### 2.4. Blood Sampling

Blood samples were collected in the morning after the subjects had fasted overnight for at least 8 h and were drawn from the antecubital area. Serum fasting glucose, aspartate aminotransferase (AST), ALT, gamma-glutamyltransferase (GGT), total cholesterol, triglycerides, high-density lipoprotein cholesterol, and low-density lipoprotein cholesterol were measured.

### 2.5. Measurements of Fatty Liver

Abdominal ultrasonography was performed by certified radiologists who were blinded to the patients’ clinical information. The sonographic diagnosis of fatty liver depended on increased liver parenchymal echogenicity compared to the adjacent kidney and spleen [31]. Since moderate and severe grades of steatosis were not sharply discriminated in many cases, fatty liver severity was divided into three groups as follows: normal, mild, and moderate-to-severe [32]. Aggravation of fatty liver was defined as the change from normal to fatty liver or from mild to higher-grade fatty liver.

### 2.6. Normality in Liver Enzymes and FIB-4

According to the conventional reference range, AST < 40 IU/L, ALT < 40 IU/L, and GGT < 45 IU/L (35 IU/L in women) were considered normal [33,34]. The FIB-4 index was calculated using the following formula: age (year) × AST (U/L))/((PLT (10^9^/L)) × (ALT (U/L))^1/2^) [35]. The value of FIB-4 < 1.45 was considered normal [35,36].

### 2.7. Statistical Analysis

The general characteristics at baseline were expressed as the means ± SD or number (proportion). The continuous and nominal variables were compared between the non-drinker and alcohol consumption groups using the *t*-test and the chi-squared test, respectively.

The change in parameters between visits 1 and 2 and visits 2 and 3 was analyzed using the paired sample *t*-test and McNemar’s test. The proportion of fatty liver development between visits 2 and 3 (the transition to the SSD) was calculated as the number of subjects with fatty liver at visit 3 divided by the number of subjects without fatty liver at visit 2. In the same way, the proportion of fatty liver development between visits 1 and 2 (the pre-SSD period) was calculated as the number of subjects with fatty liver at visit 2 divided by the number of subjects without fatty liver at visit 1. The denominator of ALT and FIB-4 abnormality was the number of subjects with ALT < 40 IU/L and FIB-4 < 1.45 at visit 1 or 2, respectively. The proportion of fatty liver development was compared between the periods from visit 1 to visit 2 and from visit 2 to visit 3 using the chi-squared test, and the risk ratio of the change between visits 2 and 3 compared to the change between visits 1 and 2 was calculated. The risk ratios were reassessed in the subgroups divided by sex, age, obesity, and smoking history. Logistic regression models were formulated to adjust for covariates. The covariates in the final model were determined according to the association between the variables and the outcomes.

All statistical analyses were conducted using the SPSS statistical package, version 26 (IBM, Armonk, Westchester, NY, USA). Results with *p*-values ≤ 0.05 were considered statistically significant.

## 3. Results

### 3.1. Characteristics of the Study Subjects

The baseline characteristics of the subjects are presented in Table 1. The mean age was 47.8 and 44.7 years in the non-drinker and alcohol consumption groups, respectively. The proportion of men was 40.6% and 69.3% in the non-drinker and alcohol consumption groups, respectively. In the alcohol consumption group, the proportion of mild fatty liver and moderate-to-severe fatty liver was 29.8% and 15.4%, respectively.

### 3.2. Changes in Liver Parameters over the Transition to the SSD Period

Table 2 shows the changes of each parameter related to liver steatosis. The proportion of alcohol consumers was not significantly different between visits 1 and 2. The response to the question regarding the drinking habit changed only in 2–3% of the study subjects. Several metabolic parameters such as diastolic BP and liver enzymes were not changed between visits 1 and 2 (the pre-SSD period), but were significantly elevated between visits 2 and 3 (transition to the SSD).

Figure 3 shows the proportion of development and aggravation of fatty liver, development of the ALT abnormality and the FIB-4 abnormality in the period between visits 2 and 3 compared to the period between visits 1 and 2. In the non-drinker group, changes in the mentioned findings were not significantly different between the periods. In contrast, in the alcohol consumption group, the development and aggravation of fatty liver and the development of ALT abnormality were significantly higher between visits 2 and 3 than in the pre-SSD period. In the alcohol consumption group, the difference in fatty liver change on ultrasonography and the development of ALT abnormality between the periods were significant in women and men, respectively (Figure 4). The proportion of abnormality in FIB-4 was not different between the periods. The differences between the periods were prominent in younger (<50 years), nonobese, and never-smoker groups.

After adjusting for the confounders (age, sex, obesity, glucose, and smoking), the development of fatty liver was significantly higher between visits 2 and 3 than in the pre-SSD period (OR = 1.300, 1.072–1.576) in the alcohol consumption group (Table 3). In the same way, the aggravation of fatty liver and the development of ALT abnormality were significantly different between the periods in the multivariate logistic regression models in the alcohol consumption group. In the models, the variables of sex, obesity, and glucose were significant risk factors for fatty liver changes. The development of ALT abnormality was substantially higher in men than in women in the multivariate model for the alcohol consumption group (OR = 3.859, 2.746–5.423). In contrast, the development of fatty liver was not different between the periods from visit 1 to visit 2 and from visit 2 to visit 3 in the non-drinker group.

## 4. Discussion

The present study evaluated the development and aggravation of fatty liver in the transition period to SSD amid the COVID-19 pandemic compared to the pre-SSD period in Korea. In the alcohol consumption group, fatty liver was more developed and aggravated in the transition period than in the pre-SSD period, especially in women. The ALT abnormality was more developed in the transition period than in the pre-SSD period, especially in men. However, the fatty liver changes were not different between the periods in the non-drinker group.

During the defined SSD period (July 2020–June 2021), the mean number of newly diagnosed patients with COVID-19 was only 394 per day [37], which was substantially lower than the number in March 2022 in Korea. Hence, although information regarding the COVID-19 infection history of the subjects is absent from our analysis, it is unlikely that the viral infection gave rise to the changes of liver parameters in our analysis. Instead, it is possible that in the SSD period, the regulations set by the Korean government contributed to lifestyle changes that may have influenced liver parameters. In December 2020, the Korean government increased the strictness of social distancing regulations [18]. This regulation recommended stay-at-home policies and restricted private gatherings (no more than five people), mass gatherings (schools, sports events, films, or musical shows), and the use of multiperson facilities, including fitness centers. This strict regulation was maintained until May 2021.

Recent studies have suggested that lifestyle changes during the COVID-19 pandemic, including a decrease in physical activity and an increase in alcohol consumption, are associated with obesity-related metabolic diseases [7,19]. Furthermore, several studies have reported that alcohol consumption increased during the pandemic [23,24], and that subjects with COVID-19-related stress drank more [23]. In a short-term analysis, increased alcohol consumption during the pandemic was shown to be associated with an increase in alcohol-associated liver disease [38]. In our study, changes in fatty liver were more pronounced in the transition period than in the pre-SSD period in alcohol consumers. Although a sedentary lifestyle, such as that often adopted during the SSD implementation, is a risk factor for NAFLD [39], obesity and metabolic syndrome may also exacerbate progression of alcoholic liver disease [40].

Alcohol intake increases the likelihood of hypertension, hypertriglyceridemia, and hyperglycemia [41]. Accordingly, our study showed that metabolic factors such as the BMI, BP, and glucose were significantly higher in the alcohol consumption group than in the non-drinker group at baseline. People with a higher BMI were likely to gain weight during the COVID-19 pandemic [42]. Since the non-drinker group was metabolically healthier and free from the hepatic impact of alcohol, the subjects in the non-drinker group may have been at less of a risk of gaining weight. However, a study conducted in Korea reported no significant differences in the BMI and the prevalence of metabolic syndrome before and after the COVID-19 pandemic [43]. Since the subjects in the non-drinker group were metabolically healthy and at a low risk of weight gain, fatty liver status may not have been influenced by SSD in these subjects.

In the alcohol consumption group, the women were found to be more vulnerable to changes in fatty liver than the men. It has been reported that women suffered from more emotional distress than men since the start of the pandemic [44]. Several studies have demonstrated that women are more commonly affected by alcohol-mediated liver disease than men [45,46]. Women are also likely to develop hepatic pathology more rapidly and to a greater extent and have higher levels of endotoxins than men [47]. In our analysis, the results showed that, compared to the women, the men had a higher likelihood of the ALT abnormality. Since the men in our analysis had higher baseline enzyme levels than the women, the elevation over a specific cutoff may have been more prominent in the men overall.

Although imaging methods were not introduced in our study to determine hepatic fibrosis, the FIB-4 equation was used as a surrogate marker. The changes in FIB-4 were not different between the periods. Considering that the subjects with viral hepatitis were excluded, the SSD period may not have been sufficiently long to detect the fibrotic change. Although genetic and environmental factors are also involved in the occurrence of hepatic fibrosis, the etiology of hepatic fibrosis remains incompletely understood. Further research is needed to determine whether hepatic fibrosis is influenced by social environments such as those that include SSD measures.

Our study has several limitations. Firstly, histological assessment was not available, which is the gold standard method for diagnosing fatty liver disease. However, we defined fatty liver using abdominal sonography which is noninvasive and commonly used to diagnose fatty liver disease. Abdominal sonography is an established screening tool with a sensitivity of 60–94% and a specificity of 66–95% [48]. Secondly, since our subjects included adults who voluntarily attended medical checkup, the subjects of this study may have been relatively healthy compared to the general population, which adds an element of selection bias to our study. Additionally, not every participating institute collected specific personal information, such as admission history. Nonetheless, the study subjects were recruited from multiple hospitals nationwide, which may have controlled for regional differences in health. Third, we lacked quantitative data regarding alcohol consumption. The alcohol consumption group may have included subjects without significant alcohol consumption. Furthermore, changes in the amount of alcohol consumption were not monitored and should be assessed in future studies. Additional studies are needed to compare the development and progression of fatty liver disease during the COVID-19 pandemic and after the COVID-19 pandemic to more fully understand the impact of SSD systems on fatty liver disease. Fourth, the integrity of laboratory data was not evaluated among the participating institutes. However, each laboratory was accredited for the accuracy of reported results, and the change of laboratory results was compared between the timepoints for each subject.

## 5. Conclusions

Our results suggest that an SSD system may influence changes in sonographic measures of fatty liver and serum levels of liver enzymes. These changes were more prominent in self-reported alcohol consumers. Sex and obesity were major factors influencing liver changes independently of chronological differences. Further research of the post-pandemic period is needed to assess long-term changes in fatty liver disease.

## Figures and Tables

**Figure 1 ijerph-19-10444-f001:**
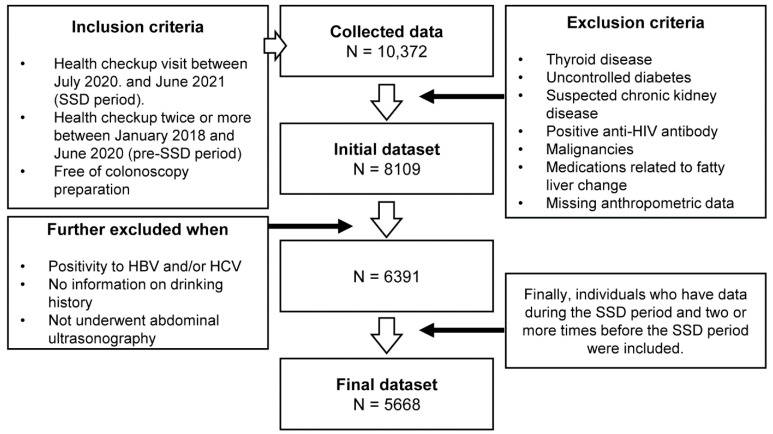
Flowchart of the study. SSD, systematic social distancing.

**Figure 2 ijerph-19-10444-f002:**
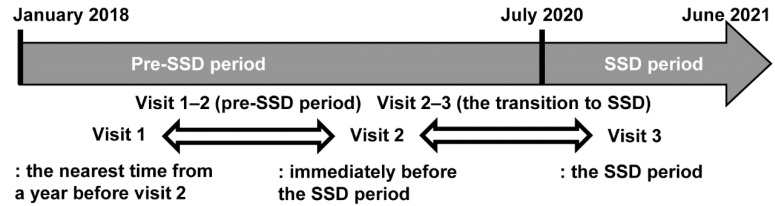
Definition of each visit. SSD, systematic social distancing.

**Figure 3 ijerph-19-10444-f003:**
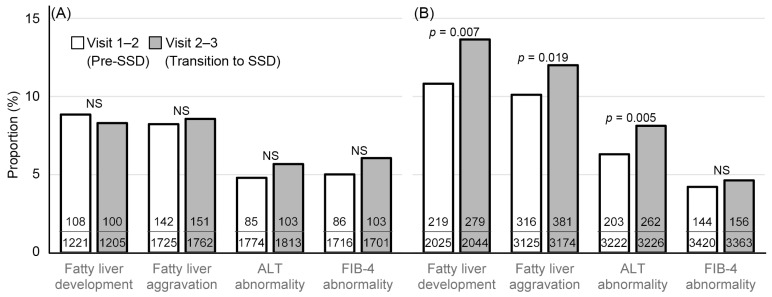
Fatty liver change comparison between the SSD transition and pre-SSD periods in (**A**) the non-drinker and (**B**) alcohol consumption groups. The proportions were compared using the chi-squared test. NS: not significant.

**Figure 4 ijerph-19-10444-f004:**
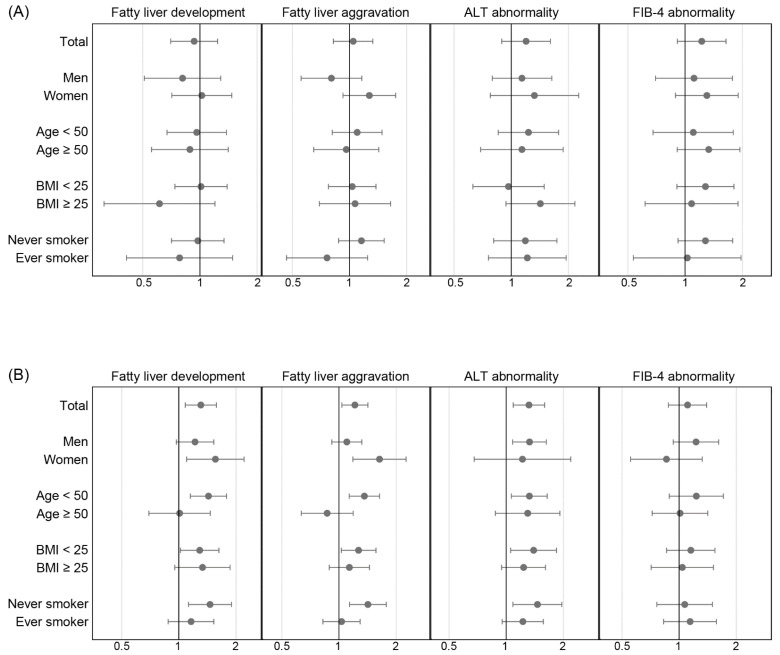
The SSD transition odds ratios compared to the pre-SSD change in (**A**) the non-drinker and (**B**) alcohol consumption groups. Subgroup analyses were conducted according to sex, age, obesity, and smoking history. Dots and error bars show odds ratios and 95%CI.

**Table 1 ijerph-19-10444-t001:** Baseline characteristics of the subjects according to alcohol history.

	Total (N = 5668)	Non-Drinker Group (N = 1972)	Alcohol Consumption Group (N = 3696)	*p*
Age	45.8 ± 8.9	47.8 ± 10.2	44.7 ± 7.9	<0.001
Sex (men)	3362 (59.3%)	800 (40.6%)	2562 (69.3%)	<0.001
Metropolitan area	4452 (78.5%)	1187 (60.2%)	3265 (88.3%)	<0.001
Hypertension	756 (13.3%)	254 (12.9%)	502 (13.6%)	0.484
Diabetes	276 (4.9%)	118 (6.0%)	158 (4.3%)	0.005
Dyslipidemia	536 (9.5%)	194 (9.8%)	342 (9.3%)	0.504
Smoking				
Non-smoker	3518 (62.1%)	1526 (77.4%)	1992 (53.9%)	<0.001
Ex-smoker	1120 (19.8%)	253 (12.8%)	867 (23.5%)	
Current smoker	1030 (18.2%)	193 (9.8%)	837 (22.6%)	
Measurements				
Systolic BP (mm Hg)	119.0 ± 14.2	118.4 ± 14.7	119.3 ± 13.9	0.037
Diastolic BP (mm Hg)	74.5 ± 10.7	73.5 ± 10.7	75.1 ± 10.6	<0.001
Pulse rate	72.1 ± 11.5	72.2 ± 11.8	72.0 ± 11.4	0.495
BMI (kg/m^2^)	23.8 ± 3.4	23.4 ± 3.5	24.0 ± 3.3	<0.001
Laboratory tests				
AST (U/L)	22.8 ± 10.7	21.7 ± 10.2	23.3 ± 10.9	<0.001
ALT (U/L)	23.9 ± 17.8	22.0 ± 17.1	24.9 ± 18.0	<0.001
GGT (U/L)	33.0 ± 36.3	24.7 ± 26.6	37.5 ± 39.9	<0.001
Glucose (mg/dL)	96.0 ± 17.9	94.8 ± 17.2	96.7 ± 18.2	<0.001
FIB-4	0.95 ± 0.46	1.01 ± 0.55	0.92 ± 0.40	<0.001
Fatty liver on sonography				
No steatosis	3246 (57.3%)	1221 (61.9%)	2025 (54.8%)	<0.001
Mild	1604 (28.3%)	504 (25.6%)	1100 (29.8%)	
Moderate-to-severe	818 (14.4%)	247 (12.5%)	571 (15.4%)	

BP, blood pressure; BMI, body mass index; AST, aspartate aminotransferase; ALT, alanine aminotransferase; GGT, gamma-glutamyltransferase.

**Table 2 ijerph-19-10444-t002:** Changes in the parameters related to liver steatosis.

	Visit 1	Visit 2	Visit 3	*p* _1–2_	*p* _2–3_
Alcohol consumer	3696 (65.2%)	3682 (65.0%)	3637 (64.2%)	0.265	<0.001
Response change					
From drinking to not drinking		2.0%	2.7%		
From not drinking to drinking		3.1%	2.8%		
Measurements					
Systolic BP (mm Hg)	119.0 ± 14.2	119.7 ± 14.4	121.0 ± 14.0	<0.001	<0.001
Diastolic BP (mm Hg)	74.5 ± 10.7	74.3 ± 11.0	75.1 ± 11.1	0.066	<0.001
BMI (kg/m^2^)	23.8 ± 3.4	23.9 ± 3.4	23.9 ± 3.4	<0.001	<0.001
AST (U/L)	22.8 ± 10.7	22.9 ± 11.0	23.4 ± 10.5	0.311	0.005
ALT (U/L)	23.9 ± 17.8	23.9 ± 18.4	24.6 ± 18.3	0.815	0.002
GGT (U/L)	33.0 ± 36.3	32.7 ± 36.8	31.5 ± 36.1	0.260	<0.001
Glucose (mg/dL)	96.0 ± 17.9	97.2 ± 18.7	97.5 ± 18.3	<0.001	0.114
FIB-4	0.95 ± 0.46	0.99 ± 0.48	1.02 ± 0.49	<0.001	<0.001

**Table 3 ijerph-19-10444-t003:** Multivariate logistic regression models for fatty liver change in the non-drinker and alcohol consumption groups.

	Fatty Liver Development		Fatty Liver Aggravation		ALT Abnormality	
	Odds Ratio	*p*	Odds Ratio	*p*	Odds Ratio	*p*
Non-drinker group						
Between visits 2 and 3 (vs. between visits 1 and 2)	0.912 (0.684–1.216)	0.529	1.034 (0.813–1.315)	0.786	1.168 (0.862–1.583)	0.317
Age (/10 years)	0.945 (0.825–1.082)	0.415	0.934 (0.829–1.052)	0.260	0.800 (0.684–0.936)	0.005
Men (vs. women)	1.617 (1.117–2.343)	0.011	1.155 (0.843–1.581)	0.369	3.137 (2.122–4.638)	<0.001
Obesity (≥25 kg/m^2^)	1.910 (1.320–2.764)	<0.001	1.646 (1.257–2.154)	<0.001	2.908 (2.121–3.986)	<0.001
Glucose (/10 mg/dL)	1.184 (1.070–1.311)	0.001	1.062 (0.985–1.145)	0.117	1.068 (0.995–1.146)	0.070
Ever-smoker	1.008 (0.643–1.578)	0.973	1.066 (0.744–1.528)	0.728	1.092 (0.755–1.580)	0.640
Alcohol consumption group						
Between visits 2 and 3 (vs. between visits 1 and 2)	1.300 (1.072–1.576)	0.008	1.212 (1.033–1.422)	0.018	1.320 (1.087–1.603)	0.005
Age (/10 years)	1.041 (0.923–1.175)	0.512	0.947 (0.854–1.050)	0.299	0.852 (0.751–0.967)	0.013
Men (vs. women)	1.355 (1.049–1.751)	0.020	1.415 (1.135–1.765)	0.002	3.859 (2.746–5.423)	<0.001
Obesity (≥25 kg/m^2^)	2.592 (2.096–3.206)	<0.001	1.846 (1.559–2.186)	<0.001	2.282 (1.872–2.782)	<0.001
Glucose (/10 mg/dL)	1.120 (1.058–1.186)	<0.001	1.041 (0.994–1.090)	0.085	1.084 (1.036–1.134)	<0.001
Ever-smoker	1.058 (0.840–1.332)	0.635	0.937 (0.778–1.128)	0.492	1.019 (0.825–1.260)	0.859

SSD, systematic social distancing period.

## Data Availability

Not applicable.

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
