# Peer review of "Fatty Liver Change in Korean Adults in a Systematic Social Distancing System Amid the COVID-19 Pandemic: A Multicenter Analysis"

_ijerph, 2022, doi:10.3390/ijerph191610444_

Round 1
Reviewer 1 Report
By virtue of this study, the authors tried to evaluate the Fatty liver changes in the Korean adults in a systematic social dis-2 tancing system amid the COVID-19 pandemics: a multi-center 3 analysis.
However the authors need to explain the following:
1. The conceptual framework is completely missing in the manuscript which is critical to this manuscript.
2. Introduction section needs improvement.
3. How did the authors addressed the potential bias issue in this study as is a retrospective study?
4. Did authors collected blood samples from the patients in this retrospective study as described by them in section 2.4?
5. Did the authors appoint certified radiologists and blinded to perform the abdominal ultrasonography?
6. Authors mentioned that," Aggravation of fatty liver was defined when the grade of fatty liver changed from normal to fatty liver or from mild to higher grade fatty liver".---Is this the definition concepttualised by the authors (or) is it a standard accepted definition?
7. What are the steps taken by the authors to maintain the data accuracy and data integrity?
8. More statistical analyses to be performed to analyse the data more intrinsically and to arrive at valid results and thereby plausible conclusions.
9. Discussion needs improvement discussing the results obtained with the evidence available in the literature.
Author Response
Than you for the grateful comments. We prepared sincere responses following each comment.
- The conceptual framework is completely missing in the manuscript which is critical to this manuscript.
Response>> Thank you for the reviewer’s comment. Although we described the background findings of the hypothesis of this study in the third paragraph of Introduction, the hypothesis was not addressed enough. We corrected the end of the Introduction as follows:
These lifestyle changes may influence liver fat accumulation. However, it remains unknown if the epidemiologic status of hepatic steatosis changed during the pandemic. It could be hypothesized that the SSD may influence the prevalence of fatty liver in the population under the regulation.
- Introduction section needs improvement.
Response>> As the reviewer commented, the hypothesis of this study was added.
- How did the authors addressed the potential bias issue in this study as is a retrospective study?
Response>> As the reviewer commented, the retrospective studies may have biases. Initially, we collected data with at least 3 time-points. It is possible that the subjects may not be a representative sample under the national regulatory condition. Therefore, we clarified the process of the inclusion and exclusion in the Study Design section. In addition, the limitation of the retrospective design is written in limitation section of Discussion as follows: relatively healthy compared to the general population.
- Did authors collected blood samples from the patients in this retrospective study as described by them in section 2.4?
Response>> Most health promotion centers have similar protocols to collect blood samples. The institutes participated in this study keep the rule of overnight fasting.
- Did the authors appoint certified radiologists and blinded to perform the abdominal ultrasonography?
Response>> Because of the retrospective design, nobody was not aware of this study at the time of ultrasonography. The certified radiologists were hired in each institute. They were not appointed by the authors but obeyed the standard to collect images and evaluate fatty liver grades. The certification of the radiologists was described in the Method section 2.5: abdominal ultrasonography was performed by certified radiologists who were blinded to the patients clinical information.
- Authors mentioned that," Aggravation of fatty liver was defined when the grade of fatty liver changed from normal to fatty liver or from mild to higher grade fatty liver".---Is this the definition concepttualised by the authors (or) is it a standard accepted definition?
Response>> As we described Method section 2.5, fatty liver severity is divided into three groups. In general, more severe grade of fatty liver contains more hepatic fat contents. Hence, the increase in the severity grade can be accepted as aggravation. Unfortunately, more severe grade of fatty liver was read as “moderate to severe” by the radiologists. In this reason, we described the fatty liver more severe than “mild” as “moderate or severe” in the whole manuscript. To avoid this misunderstanding, we added a phrase in the Method section 2.5 as follows: since moderate and severe grade of steatosis was not sharply discriminated in many cases.
- What are the steps taken by the authors to maintain the data accuracy and data integrity?
Response>> Thank you for this important comment. Because the data are provided to the examinees, the data should be reviewed before the disclosure in all institutes of this study. However, since the data of this study were collected from 13 institutes, the analytic methods may be different one another. In addition, since the comparative tests of the reagents for the blood tests were not performed, the data integrity may not be perfect. However, the anthropometric measurements and sonographic methods are standardized. Each laboratory for liver function tests was accredited by laboratory medicine foundation. We added a limitation as follows:
Fourth, the integrity of laboratory data was not evaluated among the participating institutes. However, each laboratory was accredited for the accuracy of reported results, and the change of laboratory results was compared between the time points in each subject.
- More statistical analyses to be performed to analyse the data more intrinsically and to arrive at valid results and thereby plausible conclusions.
Response>> We agree with the reviewer’s comment. We additionally analyzed the change of major variables including drinking status. We also added a result of non-drinker group in the Table 2.
- Discussion needs improvement discussing the results obtained with the evidence available in the literature.
Response>> We partly revised Discussion section according to the grateful comments by all reviewers.
Reviewer 2 Report
The authors highlight an important health issue the fatty liver; development, and aggravation due to the effect of alcohol, and the lifestyle changes during COVID-19 lockdown and social distancing policies.
The manuscript text is clearly written and well organized. The introduction and the background are reasonable given the premise of the article. The figures and tables are comprehensive and helpful. I recommend the manuscript for acceptance.
Author Response
Thank you for your favorable opinion. We revised the manuscript according to the other reviewers. We are going to try further for the improvement.
Reviewer 3 Report
The authors examined the fatty liver change before and after systmeic social distancing system during COVID-19 pandemic.
There were some comments as shown in the followings
1. The numbers of ultrasound examinations should be clearly defined before and after SSD rather than visit 1-2 and visit 2-3, that would make confusion.
2. For example once liver sono examination before SSD and once liver sono examination in SDD. Besides, the examination timing of liver ultrasound before and after SSD should also be defined. For example, 6 -12 months before SDD and 6-12 months after SSD.
3. The lifestyle habits changed before and after SSD? Importantly, the alcohol drinking habits? Exercise habits? BMI change? Laboratory test change ? Change of these data before and after SSD should be described in a new table
4. In the table 1, there should be a column to describe baseline characteristics of all participants.
5. In the table 2, why only describing alcohol consumption group? How to select the possible confounder for adjustment?
6. In the figure 3, the parameters for subgroup analysis different from those parameters in the multivariate analysis in the Table 2 . For example age>=50 and age <50, BMI>=25 and <25 in the figure 3, but in the table 2, the Age (/10 years) and obesity were used.
7. Any people got resolution of fatty liver after SSD ?
8. In the abstract , the term “transition period” mean ? transition from pre-SSD to SSD or just the period during SSD
9. The time period necessary for fatty liver development change ? Any previous papers mentioned about that
10. The subjects had any acute events during observation period, such as admission
How about effects of medications on fatty liver change?
Author Response
Thank you for the grateful comments. We tried to answer to each comment and revise the manuscript according to the review comments. Here are the responses we have prepared sincerely.
- The numbers of ultrasound examinations should be clearly defined before and after SSD rather than visit 1-2 and visit 2-3, that would make confusion.
For example once liver sono examination before SSD and once liver sono examination in SDD. Besides, the examination timing of liver ultrasound before and after SSD should also be defined. For example, 6 -12 months before SDD and 6-12 months after SSD.
Response>> We agree with the reviewer’s comment. The term of visit 1 to 3 might make confusion. We added a figure to explain each visit point. In addition, further explanation is added in Figure 2 and Table 2.
- The lifestyle habits changed before and after SSD? Importantly, the alcohol drinking habits? Exercise habits? BMI change? Laboratory test change ? Change of these data before and after SSD should be described in a new table
Response>> Thank you for the comment. Further explanation was needed to show the changes between the time points. Small proportion of the subjects (2–3%) changed their response to the question asking drinking habit.
- In the table 1, there should be a column to describe baseline characteristics of all participants.
Response>> We agree with the comment. We added a column in Table 1.
- In the table 2, why only describing alcohol consumption group? How to select the possible confounder for adjustment?
Response>> Thank you for the comment. Because non-drinker group does not show a significance in Figure 2 and 3, we have described only alcohol consumer group. Now, we added several rows for non-drinker group in Table 2. Additionally, we collected various variables that may potentially influence fatty change. Among the potential confounders described in Table 1, the variables that were not associated with the outcomes in univariate analyses were not included in the final analyses. We added a sentence in the statistical analysis section as follows: the covariates in the final model were determined according to the association between the variables and the outcomes.
- In the figure 3, the parameters for subgroup analysis different from those parameters in the multivariate analysis in the Table 2 . For example age>=50 and age <50, BMI>=25 and <25 in the figure 3, but in the table 2, the Age (/10 years) and obesity were used.
Response>> Thank you for the comment. First, obesity is defined as BMI ≥ 25 in Korean population. We added this definition in Method 2.3 Measurement section. We also added an explanation in Table 2.
In Figure 3, we intended to show the results according to age subgroup. The results were not changed when using two age groups (50 years) instead of 10 years gap. The results are shown as follows:
Original:
Visit 1.300 (1.072-1.576) 0.008 1.212 (1.033-1.422) 0.018 1.320 (1.087-1.603) 0.005
Age10 1.041 (0.923-1.175) 0.512 0.947 (0.854-1.050) 0.299 0.852 (0.751-0.967) 0.013
Using age groups divided by 50 years:
Visit 1.302 (1.074-1.579) 0.007 1.209 (1.030-1.418) 0.020 1.314 (1.082-1.596) 0.006
Age50 1.062 (0.852-1.325) 0.590 0.937 (0.781-1.124) 0.482 0.761 (0.612-0.947) 0.015
- Any people got resolution of fatty liver after SSD ?
Response>> Thank you for the comment. We have analyzed this point. However, since this study aimed to show the development and aggravation of fatty liver-related factors, we did not show the results. The results were as follows:
Non-drinker: 0.920 (0.709–1.194), P=0.576
Alcohol consumer: 0.698 (0.588–0.829), P<0.001
It is interpreted that fatty liver was less improved in the SSD-transition period than in the pre-SSD period in alcohol consumers. Hence, the data is consistent with the study concept. Please understand the intention not to show this result.
- In the abstract , the term “transition period” mean ? transition from pre-SSD to SSD or just the period during SSD
Response>> We agree that the term may make confusion. However, in the 4th line of the abstract, the transition period (pre-SSD to SSD) was defined. Please understand that word count limitation in the abstract hinders further explanation of the contents.
- The time period necessary for fatty liver development change ? Any previous papers mentioned about that
Response>> Theoretically, alcohol intake can lead to a build-up of fats in the liver, and liver damage may be reversed after at least 2-week abstinence of alcohol. We have found a study which followed up fatty liver disease prevalence in an eastern Chinese community for 11 years (Reference: BMJ Open. 2022 Jun 27;12(6):e054891, PMID: 35760549). It can be concluded that fatty liver changes within a year. The study subjects in our study were followed up with about 1–2-year interval.
- The subjects had any acute events during observation period, such as admission
Response>> Thank you for the comment. Unfortunately, we did not collect the admission history. Specific status to require admission may affect fatty liver changes. At least, we excluded the subjects with potential disease to admit. We added a sentence in the limitation section as follows: additionally, not every participating institute collected specific personal information like admission history.
- How about effects of medications on fatty liver change?
Response>> As we described in Method 2.1, long-term use of medications related with fatty liver change was excluded. Despite the exclusion, 5–10% of the subjects were taking medications for hypertension, diabetes, and dyslipidemia. Further analyses revealed that any of medication history was not significantly associated with fatty liver-related factors (not shown in the manuscript).
Round 2
Reviewer 1 Report
Authors have addressed all the queries as raised by the reviewers adequately
Reviewer 3 Report
The authors had made responses to my questions.